

# A review of social background profiling of speakers from speech accents

Mohammad Ali Humayun[1], Junaid Shuja[2] and Pg Emeroylariffion Abas[3]

[1] Department of Computer Science, Information Technology University, Lahore, Pakistan
[2] Department of Computer and Information Sciences, Universiti Teknologi PETRONAS, Seri Iskandar, Malaysia
[3] Faculty of Integrated Technologies, Universiti Brunei Darussalam, Jalan Tungku Link, Brunei

## ABSTRACT

Social background profiling of speakers is heavily used in areas, such as, speech forensics, and tuning speech recognition for accuracy improvement. This article provides a survey of recent research in speaker background profiling in terms of accent classification and analyses the datasets, speech features, and classification models used for the classification tasks. The aim is to provide a comprehensive overview of recent research related to speaker background profiling and to present a comparative analysis of the achieved performance measures. Comprehensive descriptions of the datasets, speech features, and classification models used in recent research for accent classification have been presented, with a comparative analysis made on the performance measures of the different methods. This analysis provides insights into the strengths and weaknesses of the different methods for accent classification. Subsequently, research gaps have been identified, which serve as a useful resource for researchers looking to advance the field.

## INTRODUCTION

Speaker profiling is the process of estimating the characteristics of an unknown speaker using a classification model that has no prior instance of speeches from the speaker to be profiled. Identification of a speaker's social origin considering the speech accent characteristics is termed as social background profiling of speakers. The social background of speakers heavily influences their accents in pronouncing different phonemes and the pattern of using them in their speech. These involuntary characteristics in human speech can be used to identify the speaker's social background without the need for the speaker to reveal it explicitly (*Kalluri, Vijayasenan & Ganapathy, 2020*; *Singh, Raj & Baker, 2016*). Speech accent classification considers various linguistic and prosodic features, such as pronunciation, rhythm, intonation, and stress patterns, to determine the accent of a speaker.

Social background profiling from speech has multiple applications, and much research on speaker background profiling has been carried out by people from different disciplines with different objectives. The most widely-used background profiling application is for adapting automatic speech recognition (ASR) models. ASR models perform relatively poorly for accented speech. However, speakers with different backgrounds commonly have

Corresponding author
Mohammad Ali Humayun,
ali.humayun@itu.edu.pk

a relatively large variation in their accents, especially when speaking a second language. Hence, accent identification has been incorporated into ASR to reduce the error rate. Multiple models are normally trained for different accents, and models trained particularly for a particular accent in the test speech are used for ASR after the classification of the speaker's accent. Performance results of accent-specific ASR models can highlight the accents that are difficult to recognise; hence acquiring more training data for such accents can lead to more robust ASR (*Najafian & Russell, 2020*; *Weninger et al., 2019*).

Social background profiling is also used for the forensic investigation of criminals by utilising a piece of speech evidence related to a crime (*Jessen, 2007*). As opposed to speaker verification or identification, which is used as forensic evidence to match the identity of a convict, speaker profiling can assist investigators in tracking down an unknown criminal by locating him or her to belong to a specific geographical location or ethnic group. However, the forensic application of speaker profiling has some particular challenges. The difference between accents to be classified may be marginal, which makes it difficult to associate a criminal with a precise geographical location within a country. Furthermore, speech data available for forensic profiling from crime-related evidence are commonly, of very small duration and low quality. More importantly, the margin of error and reliability of the classification are of critical importance as the output of the classification model has legal implications and can be used as a basis for conviction.

Forensic speaker profiling research has focused on the impact of proximate accents as well as linguistic content on accent classification accuracy. Research has also focused on identifying particular phonemes, which are more useful for the classification of particular accents (*Brown & Wormald, 2017*). The quality and duration of the available samples for forensic analysis are also critical, as mostly the evidence is from a telephone call, with the telephone channel acting as a bandpass filter; shifting the formant frequencies for vowels which are crucial for the classification task (*Kunzel, 2001*; *Moreno & Stern, 1994*). Additionally, the commonly short duration of test samples results in the availability of fewer vowels in the speech, which are also critical for the classification task (*Brown, 2018*). Social, linguistics, forensic, and speech communities have emphasised the need for collaborative research in forensic speech science to avoid innocent convictions and discrimination towards a particular community during the course of criminal investigations or justice processes (*Hughes & Wormald, 2020*).

Speaker social background profiling can also be used in automated customer services and call centers for identifying the background of a speaker and serving them accordingly. It can also be used for targeted marketing based on the speaker's background. Finally, it can be used in sociolinguistic and psycholinguistic research to analyze the differences in accent across different societies and geographic regions.

Although significant research works have proposed speaker social background profiling models, no review work has surveyed the state of the art in this area. Some surveys have targeted speaker profiling in general, but none is dedicated specifically to the social backgrounds of speakers. This work surveys recent research on automatic social background profiling for English language speech. The survey covers various aspects of social profiling, including accent classification, dialect identification, and native language identification.

The survey examines the application scenarios targeted by the studies and the corresponding challenges.

## RATIONALE AND AUDIENCE

Automatic social background profiling of speakers has numerous practical applications, such as in the development of multilingual speech recognition systems, the creation of personalized speech technologies, and the analysis of speech data for sociolinguistic research. Moreover, recent advancements in machine learning techniques such as deep learning, have led to significant improvement in the accuracy and robustness of social background classification systems.

Despite the significance and rapid advancements in speaker background or accent profiling research, an exhaustive literature review is missing dedicated particularly to speech accent classification. This literature review aims to fill that gap by comprehensively examining recent research regarding speech accent profiling in the context of accents in the English language. Given the diverse applications of speaker accent classification across various domains, this review article caters to a broad audience, encompassing the speech recognition research community, forensic science, and sociolinguistic communities.

This article presents an overview of the current state of research in speech accent classification, including the datasets, speech features and classification models used, and the performance measures achieved. The strengths and weaknesses of the different methods were highlighted. A comparison between and evaluation of different studies, as well as the research gaps that need to be addressed to advance the field, were also made.

This article reviews the recent research on speaker background profiling from speech accents in the English language. The contributions of the article are as follows.

1. It formulates a detailed taxonomy for speaker accent profiling in terms of speech features and the machine learning models used for classification.
2. It provides a detailed literature review of speaker profiling models proposed in recent studies.
3. It provides a comparative analysis of the literature review to present the strengths and weaknesses of the research.
4. It lists research challenges and future research directions to serve as a resource for researchers looking to advance the research on speaker profiling.

The rest of the article is organised as follows. The survey methodology adopted in this article is given in the next section. The following section presents the taxonomy of speech features and machine learning models used for accent classification. The section next reviews recent literature for accent profiling including the datasets designed, the features used and classification models applied for the speaker background profiling tasks. Then a comparative analysis section provides comparisons between state-of-the-art models for speaker background profiling. The next section presents lays down future research directions in the area and the last section presents the conclusion.

# SURVEY METHODOLOGY

This literature survey systematically evaluates the current state of research in this speech accent classification. It identifies the key research issues and evaluates the effectiveness of different methods. The survey highlights commonly used datasets and performance metrics used for the evaluation of the proposed strategies. Finally, the survey points out the opportunities for future research in the area.

The survey has focused on peer-reviewed journal articles and conference proceedings published within the recent years. The search filtered English language publications published after 2013. The keyword combination used for search was ''automatic English speech speaker profiling'' OR ''native language'' OR dialect OR accent OR ''social background''. The search targeted all papers containing each of the terms 'automatic', 'English' 'speech', and 'speaker' throughout the manuscript and required them to include any of the following: profiling, native language, dialect, accent, or social background which yielded 17,900 results. The results were then filtered, selecting datasets with at least 100 speakers, 2,000 speech samples, and five categories, except for the Accent and Identity of Scottish English Border (AISEB) dataset, which was kept due to its significance for forensic and sociolinguistic applications. The AISEB dataset records accent variations across Scottish English border towns with similar accents. After filtering, nine datasets were analyzed for their state-of-the-art methodologies and results, resulting in a detailed analysis of 10 landmark studies that utilized the shortlisted datasets.

Speech accent classification models have used a variety of datasets for the English language with speaker background annotations. Most models transform these speech samples to derive suitable features for classification and then feed them to various classifiers for the classification tasks. The following information was extracted from each article:

- Speech features
- Research methods and techniques
- Evaluation metrics and datasets
- Results and conclusions
- Limitations and future research opportunities

The extracted information was analyzed and organized to present the key trends in speech accent classification research. The main strengths and limitations of current methods have also been compared.

## TAXONOMY

This section focuses on categorizing the speaker profiling research based on speech accent features and classification models. Figure 1 summarizes the types of features extracted from speech and classification models used for background profiling.

### Features

The speech features can be generally divided into prosodic, short-term, long-term and phonotactic features.

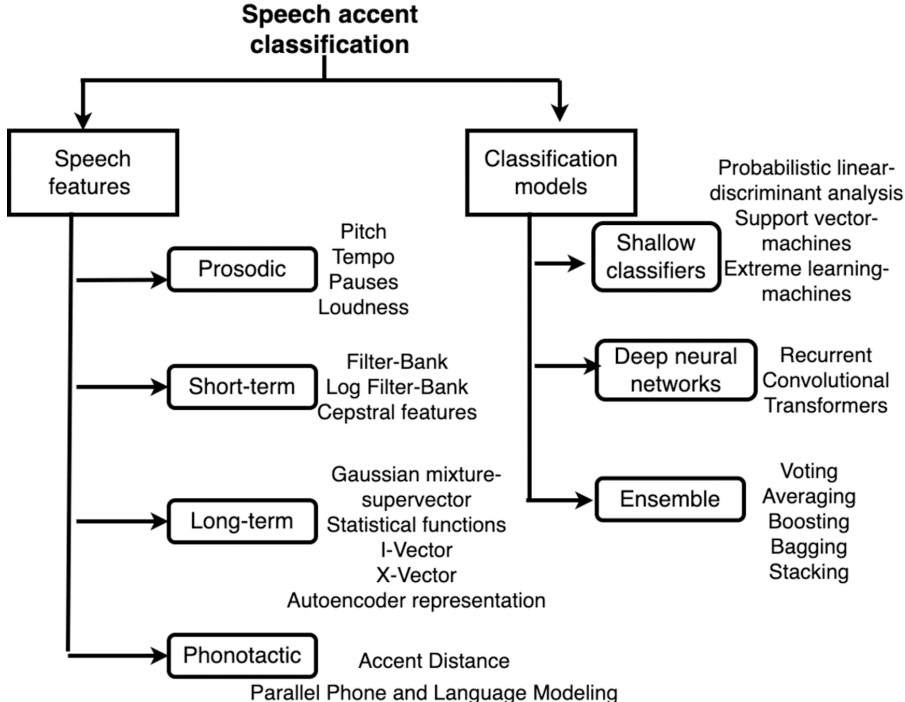

**Figure 1** Taxonomy of features and models for accent classification.

## Prosodic features

The prosodic features are independent of speech content and include parameters such as pitch, volume, stress levels, pauses, and tempo. These features are represented by parameters such as fundamental frequency, average intensity and speed of variations in the acoustic signal (*Jiao et al., 2016a*).

## Short-term features

Short-term (ST) features are extracted by small-sized quasi-stationary windows sliding across the speech duration. There are usually spectral or cepstral components for short-duration chunks of speech. Since speech is a time-varying signal produced by changing shapes of the vocal tract as a system response to the glottal excitation signal, the quasi-stationary time windows are usually only a few milliseconds long as humans cannot change the vocal tract shape in time duration, less than that. Short-term features can be generally divided into spectral and cepstral features. The spectral and cepstral features extracted from the short-time windows are referred to as short-term spectral and cepstral features, respectively.

### Spectral features

The most well-known ST spectral representation used is the spectrogram, which is the sequence of spectra in the small windows plotted across time. Figure 2 illustrates an audio speech waveform with a corresponding spectrogram plotted against time.

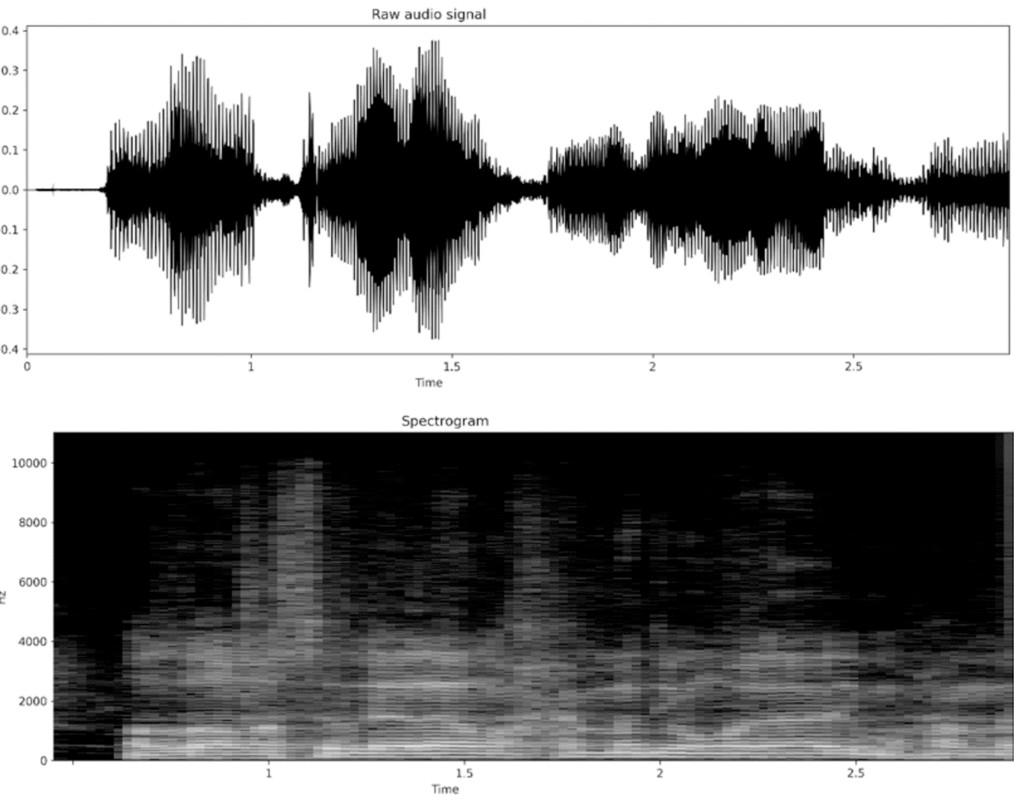

**Figure 2** **Speech waveform and its spectrogram.** Upper plot shows waveform amplitude variations over time while lower plot displays frequency content over time for the corresponding spectrogram.

The spectrogram is usually divided into a discrete bank of filters on the Mel scale, and the energies within each filter are summed to get a sequence of Filter-Bank energies. The human neurons respond to the intensity of sound on a logarithmic scale. Inspired by the logarithmic response to sound amplitude, the magnitudes of Filter-Bank energies are also converted to logarithmic representation to obtain the LOG Filter-Bank features (*Humayun, Yassin & Abas, 2023*; *Shon, Ali & Glass, 2018*).

### Cepstral features

Since speech is produced by the convolution of vocal tract response with glottal excitation, the spectrogram represents the multiplication of both responses, which are changed to additive by taking the logarithm of the spectra. The additive vocal tract and glottal components in the log of spectra are segregated by inverse frequency transform, with the resulting representation referred to as the Cepstrum. Hence, the lower Cepstrum can extract the envelope of the spectrum. The cepstral coefficients capture the spectral envelope of the speech in quasi-stationary short durations. These coefficients have significantly lower dimensions as compared to the spectrum and have very little correlation between themselves, making them suitable as input for the classification models. The most well-known variant cepstral coefficients used in literature is Mel Frequency Cepstral Coefficients (MFCC). MFCC are coefficients of cosine transform computed from energies within band

pass filters applied to mel-scaled spectrogram of speech (_Lalitha et al., 2015_; _Najnin & Banerjee, 2019_).

### Linear prediction coefficients

Linear predictive coefficients (LPC) represent the coefficients of an all-pole autoregressive model estimating the vocal tract response for a smaller time window (_Vestman et al., 2018_). The coefficients are computed by minimising the mean square of difference between actual and linearly predicted values for each sample using the coefficients. The coefficients represent the vocal tract shape whilst the difference between prediction and the actual sample values represents the glottal excitation component of speech.

## Delta features

To capture the temporal structure in the short-term features, short term features are usually appended with Delta features, where delta features for each time frame represent the local slope of coefficients over a small number of neighbouring time frames found by least-squares (_Rajpal et al., 2016_).

## Long-term embeddings

Long-term embeddings are fix-sized representations for the complete speech utterances obtained by temporal modelling of a sequence of short-term features. These long-term embeddings are particularly useful for utterance level, paralinguistic classification tasks, such as speaker identity, emotion, gender, language, accent, or dialect identification.

Long-term (LT) acoustic features contain information on the temporal structure within the complete speech utterances and are a higher dimensional statistical description of ST features across the complete speech utterance. It is particularly useful for paralinguistic information processing as it does not account for the local acoustic variations, which are more relevant for the speech-to-text translation, and hence, the main focus of paralinguistic classification models is on the long-term embeddings of the speech utterance.

There are three main methods used in literature for the LT feature vector extraction: (1) the functional description vector, (2) the parametric super vector, and (3) neural network embeddings, which have recently become popular mainly for speech and speaker recognition tasks.

### Functional description vector

The functional descriptor vector is obtained by applying multiple statistical functions across time for all the ST features and then concatenating the results. The concatenated vector has dimensions equal to the number of ST features multiplied by the number of statistical functions employed (_Kalluri, Vijayasenan & Ganapathy, 2020_).

### Parametric super vector

The long-term vector can also be constituted from parameters for the underlying probability distribution. Gaussian mixture models (GMMs) are used to model the short-term features. The parameters estimated using the short-term features for the utterance are concatenated to obtain the long-term representation (_Kinnunen & Li, 2010_; _Sethu et al., 2013_).

A long-term representation that has achieved state-of-the-art results in speaker, language, and accent recognition models is the parametric I-vector. It is obtained by factor analysis decomposition of the difference between the adapted model and Universal Background model (UBM) for Gaussian Mixture model (GMM) parameters (*Dehak et al., 2011*).

### Neural network embedding

More recently, long-term temporal modelling of ST features by neural networks has shown promising results in replacing the i-vectors. Neural networks are hierarchial parametric functions trained to minimise an optimisation loss function. The parameters of the specific bottleneck layer can be used to represent the long-term representation for a sequence of short-term features used to train a neural network for a relevant sentence classification task.

### Unsupervised representations

Auto-encoders can capture bottleneck representations of speech using unlabeled datasets. The auto-encoder comprises an encoder and a decoder part. The encoder transforms speech into the bottleneck representation (*Renshaw, 2016*). Many enhancements have been proposed for the autoencoder architecture to learn the representations for customized requirements (*Goodfellow et al., 2020*; *Gregor et al., 2014*; *Kingma & Welling, 2013*).

## Phonotactic features

Phonotactic features refer to phoneme inventory and phoneme sequence used by a speaker. Besides the phonotactic features, more complex linguistic features, including lexical, semantic, and contextual features can also be used to identify speaker backgrounds by spoken dialect in the case of spontaneous speech. The linguistic features are extracted from the transcript of a speech, obtained by a speech recognition model. However, these linguistic features are not applicable in the case of short-duration or scripted speech samples, particularly relevant for forensic profiling.

## Feature normalisation and data augmentation

Feature normalisation and input data augmentation are two well-known strategies to make a classification model more robust to irrelevant variations. The former refers to the pre-processing techniques applied to the input features to remove irrelevant variability from the data. Feature normalisation techniques include spectral subtraction, cepstral normalisation, and frequency warping. Spectral subtraction aims to remove the additive noise from a speech. The simplest method for spectral subtraction is to subtract the spectral mean of silence segments in a speech from the entire utterance (*Vincent, Virtanen & Gannot, 2018*). On the other hand, cepstral normalisation is used to counter the response of slow-changing linear channels, usually achieved by subtracting the temporal mean across a small-duration speech utterance from each cepstral coefficient (*Liu et al., 1993*). Finally, frequency warping is used to counter the speaker or corresponding vocal-tract variability. This is achieved by compressing, expanding, or rescaling the frequency axis for the spectrogram, with the warping factor estimated by training for each speaker (*Pelecanos & Sridharan, 2001*).

Contrastingly, data augmentation is the process of deliberately adding irrelevant variability in the features to augment the input training data, with the addition of augmented data for training with the same classification labels making the classification model insensitive to the irrelevant variations. Many augmentation techniques have been proposed and tested to be successful, mainly for speech and speaker recognition tasks. The most well-known speech augmentation techniques are vocal tract length perturbation (VTLP), reverberation, spectral augmentation, pitch-shifting, denoising, and speed modification.

VTLP is achieved by warping the frequency axis through multiplication with randomly selected warping factors (*Jaitly & Hinton, 2013*), whilst reverberation is achieved by convolution of the speech with recorded audio impulse response for a reverberant environment (*Snyder et al., 2018*). Spectral augmentation has been recently proposed for ASR and has been proven to be very effective (*Park et al., 2019*). It refers to randomly masking or warping the spectrogram across time as well as frequency axes. Pitch shifting simply rolls the fundamental frequency across the frequency axes. Denoising adds random noise to speech, and finally, speed modification is achieved by resampling the speech samples (*Fukuda et al., 2018*).

## Classification models

Both shallow and deep machine learning architectures have been employed in recent research as classifiers for accent profiling models. Shallow classifiers usually have a global optimisation and are more effective for simpler patterns of handcrafted features with fewer data. On the other hand, deep classifiers are hierarchical and are optimised iteratively based on error gradient. Generally, deep classifiers are more useful for complex patterns in a huge amount of data.

## Shallow machine learning models

Shallow classifiers include naïve Bayes, decision trees, K-nearest neighbours (KNN), support vector machines (SVM), and extreme learning machines (ELM), amongst many others. Inputs for the shallow models are manually crafted features that present the data as a compact representation and maximise the segregation between different classes. For speech classification tasks, the inputs are usually the statistically modelled acoustic feature vectors representing complete utterance, while for language classification tasks, the inputs are dense vector representations for sentences.

The naïve Bayes classifier estimates the posterior probability of the target class conditioned on the observed features using the Bayes rule for conditional probability. It is termed naïve as it assumes the feature probability densities conditioned for the classes to be independent of each other and hence computes their joint probability as products of their individual conditional probabilities. However, the naïve Bayes model may not be the most accurate or suitable for complex speech datasets. Other machine learning models, such as support vector machines or deep neural networks, may provide better but may also require more computational resources and longer training times. Decision trees comprise hierarchical decisions for classes based on threshold values for input features thresholds. Multiple decision trees merged for a single problem enhance the results and are called

random forests (*Schonlau & Zou, 2020*). KNN is a nonparametric classifier that stores the complete training data and decides the class for the test sample by majority voting amongst classes for k-nearest neighbours for the test sample in the training data.

Probabilistic linear discriminant analysis (PLDA) models the training data to be generated from a mixture of Gaussian distributions and classifies the samples based on latent distance from distribution centres. SVM estimates a linear hyper-plane in the feature space separating the data points from both categories; the hyper-plane is found by trying to maximise the distance between the points from each category which are the closest to the opposing category. These data points are referred to as the margin points or the supporting vector. Moreover, if the data features are not linearly separable in their original dimensions, a kernel function is used over the feature vectors to append the result as a new dimension to each vector. Consequently, the features are projected to higher dimensions which makes them linearly separable for the hyper-plane. Finally, ELM is a single-layer feed-forward neural network with randomly assigned sizes and input weights on the hidden layer. Only the output weights for the hidden layer are found using training data by linear optimisation.

## Deep learning models

Convolutional neural networks (CNNs) and recurrent neural networks (RNNs) are usually employed as deep architectures for modelling speech sequences (*Chung & Glass, 2018*; *Kim et al., 2023*; *Wang et al., 2019*). RNNs are designed specifically for time-series data, whereby input to each RNN unit is the input feature for that time step concatenated with the weighted output by the same RNN unit from the previous time steps. The final output of the RNN can be the output by the units at the last time step as well as the sequence of outputs for the entire time series. Attention is a mechanism proposed for the RNNs and is state-of-the-art for classification in most speech processing tasks (*Qian et al., 2019*; *Ubale et al., 2019*). To employ attention in RNNs, outputs for all-time steps by a single RNN unit are collapsed by weighted averaging while the weights are learned automatically during training. Consequently, critical segments in the input series are highlighted.

On the other hand, CNNs slide multiple matrices (filters), with different weights and fixed-width (kernel size), across the input series. Each convolutional layer is usually followed by a pooling layer, which locally samples by a fixed ratio from the sequence of filter outputs. Finally, global average pooling averages the output sequence from each filter to a single value. Recently, a variant of global average pooling, *i.e.,* attentive pooling, has been proposed for speech accent classification tasks (*Ubale et al., 2019*). Attentive pooling is a weighted global average with weights learned by training, with the weights for attentive pooling highlighting important input segments. Figure 3 depicts a convolution operation with two filters sliding across time for a speech spectrogram.

Convolutional layers are often followed by activation functions that transform the result of the convolution into a non-linear representation. Activation functions introduce non-linearity into the network and enable it to learn complex representations. Common activation functions used in CNNs include rectified linear unit (ReLU), sigmoid, and tanh. Max pooling and average pooling are two common types of pooling operations that reduce

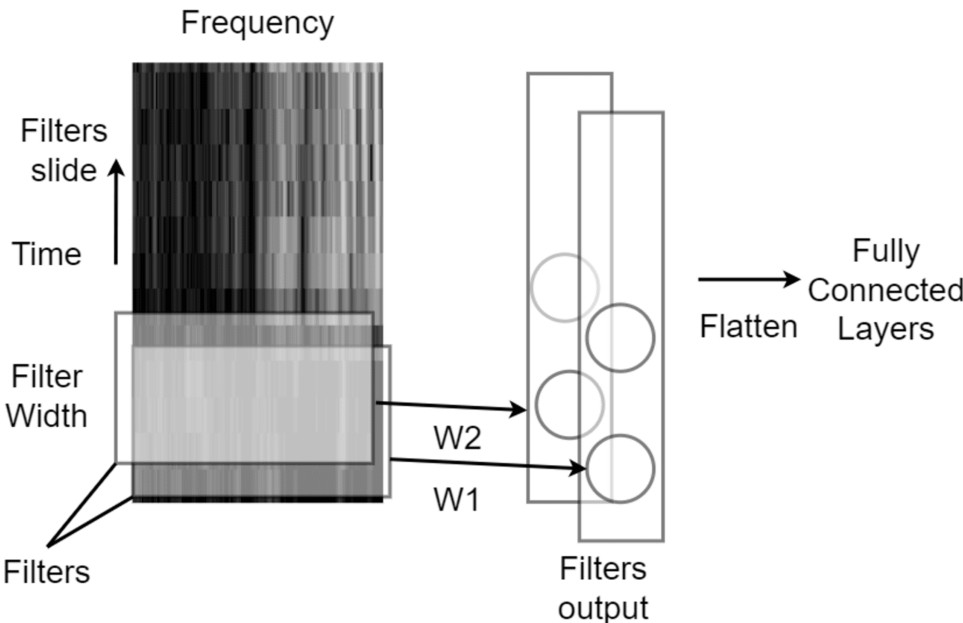

**Figure 3** Single dimensional convolutional neural network with convolutional kernel sliding across the time axis of time-frequency features.

the spatial size of the feature maps produced by the convolutional layer; Max pooling selects the maximum value from a set of adjacent activations, while average pooling takes the average of the same set of activations. These pooling operations help to make the network more robust to translation and scale variations in the input data, as well as reduce computational complexity.

In recent years, speech processing models have used transformer-based models. Transformers are capable of modelling longer dependencies between speech frames, allowing them to better capture the context of speech signals (*Vaswani et al., 2017*). It utilizes the self-attention mechanisms and originally uses an encoder–decoder architecture, to differentially weight the significance of different parts of the input data.

In terms of computation complexity, CNNs and RNNs are more computationally complex compared to SVM and ELM, as they involve multiple layers and require more computation for each layer. LDA, being a linear classifier, is computationally least complex, but it may not perform well on complex datasets.

## Ensemble of models

Utilising an ensemble of multiple classifiers has also been used to enhance the overall classification performance. Ensembling of multiple classifiers refers to the fusion of multiple classifiers for a particular task. Training multiple classifiers on the same dataset and averaging their outputs for prediction increase the classification accuracy due to the randomness in neural network training. Moreover, carefully designed variations in classifier design can add diversity to the ensemble-based model, significantly improving the model's generalisation capability (*Dong et al., 2020*).

Besides simple ensemble methods like majority voting and output averaging, advanced techniques, such as bagging, boosting, and stacking, have also been used to merge multiple individual classifiers (*Dong et al., 2020*; *Pintelas & Livieris, 2020*). Bagging is an abbreviation for bootstrapping and aggregation. Bootstrapping refers to the training of individual classifiers using multiple subsets of the dataset, where a different subset of input features might be used for each training subset. Aggregation is the process of combining the output of individual trained classifiers. On the other hand, boosting methods train the individual classifiers sequentially so that the data misclassified by the earlier classifiers is emphasised in succeeding classifiers. Data points that produce training errors in initial classifiers are identified, and the following classifiers are adjusted to minimise the training error for the misclassified data. Stacking refers to the merging of individual classifiers using a meta-learner or a meta-classifier. The meta-classifier is stacked on top of the individual classifiers to be fused such that the outputs from individual classifiers are treated as inputs for the meta-classifier. The meta-classifier fusing the individual classifiers is trained by end-to-end training.

## LITERATURE REVIEW

### Datasets

Numerous speech datasets have been compiled for research related to multiple speech-processing tasks. However, only the datasets comprising speakers from diverse backgrounds and their corresponding labelling can be used for social background classification. The datasets used in accent classification literature have speech samples labelled with the speaker's accents, in terms of their dialect, native language, or even geographic origin. These datasets can have various sizes in terms of speakers and their speech samples. The spread of accents of these datasets can also vary from global accents to regional accents within countries such as Great Britain, America, and India. Moreover, the quality of datasets can also vary, with some in a controlled setting and some crowd-sourced. Some of them are available publicly, while access to others is restricted.

The Accent and Identity on Scottish English Border (AISEB) dataset consists of English speech in four different accents from Scottish-English Border towns. Social science researchers collected the data from British Borders (BB). The Accents of British Isles (ABI) dataset contains 14 different accents of British English, whilst the Test of English as a Foreign Language (TOEFL) dataset comprises English speech samples by TOEFL speaking test candidates from across the globe and has been compiled by Education Testing Service (ETS) for the Native Language Sub-Challenge (NLSC) (*Schuller et al., 2016*). The Texas Instruments, Massachusetts Institute of Technology (TIMIT) dataset contains eight different American English accents. Common Voice is a crowd sourced (CS) dataset by Mozilla. The dataset has speech samples in 96 different languages, and English samples have been labelled with the native language of speakers from across the world.

Speech Accents Archive (SAA) is an open-source speech dataset for analysing accent variations within speeches. The dataset comprises a single sentence spoken from the written transcript by volunteer speakers spread across the world (*Weinberger & Kunath, 2011*).

**Table 1 English speech accent datasets.**

| Reference | Dataset | Accents | Spread | Speakers | Samples | Quality | Availability | Owner |
|---|---|---|---|---|---|---|---|---|
| *Garofolo et al. (1992)* | TIMIT | 8 | US | 630 | 6,300 | Controlled | Public | MIT |
| *Weinberger & Kunath (2011)* | SAA | 200 | Globe | 2,138 | 2,138 | Crowd-sourced | Public | GMU |
| *Schuller et al. (2016)* | TOEFL | 11 | Globe | 11,000 | 11,000 | Controlled | Private | ETS |
| *Ge (2015)* | FAE | 23 | Globe | 4,925 | 4,925 | Controlled | Private | CSLU |
| *Ardila et al. (2019)* | CV | 16 | Globe | 66,173 | 66,173 | Crowd sourced | Public | Mozilla |
| *West (2013)* | AISEB | 4 | BB | 160 | 160 | Controlled | Private | York |
| *Ferragne & Pellegrino (2010)* | ABI | 14 | Britain | 285 | 57 hrs | Controlled | Private | UoB |
| *Kalluri, Vijayasenan & Ganapathy (2020)* | NISP | 5 | India | 345 | 15,000 | Controlled | Public | N-I |
| *Demirsahin et al. (2020)* | OBI | 5 | Britain | 120 | 17,877 | Crowd sourced | Public | ELRA |

Two rather new datasets: NITK-IISc (NI) Speaker Profiling (NISP) (*Kalluri, Vijayasenan & Ganapathy, 2020*) and Open British Isles OBI (*Demirsahin et al., 2020*), have still not been tested for accent classification, although they have a considerable number of samples, background diversity, and accent annotations. NISP has English and regional languages by speakers from India, whilst the OBI dataset has samples with various English accents. Table 1 summarises the English datasets with accent annotations mostly used by the state-of-the-art models.

## Features

Short term Filter-Bank energies have been proven effective for several speech accent classification tasks (*Rajpal et al., 2016*; *Sailor & Patil, 2016*; *Shon, Ali & Glass, 2018*). Besides, MFCC coefficients have little correlation, which makes them suitable as input for classification models. It has been shown that MFCC features perform well for segregating accent-related information (*Singh, Pillay & Jembere, 2020*).

Based on the architecture, supervised neural network embeddings have been named according to the proposed models, such as bottle neck features (BNFs) or x-vectors (*Snyder et al., 2018*). The BNF is the representation from the penultimate layer of a neural network trained for ASR, whilst the x-vector is the representation of the intermediate layers of a neural network trained for speaker recognition. X-vectors (*Snyder et al., 2018*) are obtained by temporal pooling across the short-term features within a neural network trained for speaker discrimination.

The distance matrix between the pronunciations of different phonemes in terms of their acoustic pronunciations (*Brown & Wormald, 2017*) has proved effective for accent classification. Accent-Distance (ACC-DIST) based model utilises the distance matrix amongst the acoustic features for different spoken phonemes and has been successfully applied to identify spoken accents with slight geographic separation (*Brown, 2018*). Similarly, Parallel Phone Recognition and Language Modeling (PPRLM) models effectively recognise the spoken phonemes and use the sequence and frequency of phonemes to identify spoken accents or language dialects (*Najafian & Russell, 2020*). Table 2 summarises the speech features used in recent literature for accent classification with their types and brief descriptions.

**Table 2   Speech features for accent classification.**

| Reference | Feature | Type | Description |
|---|---|---|---|
| *Rajpal et al. (2016)* | Filter-Bank | Short-term spectral | Total energies in spectral filter applied on mel-scale |
| *Shon, Ali & Glass (2018)* | Log Filter-Bank | Short-term spectral | Logarithmic magnitude of Filter-Bank energies |
| *Singh, Pillay & Jembere (2020)* | MFCC | Short-term cepstral | Discrete cosine transform of Log Filter-Bank |
| *Rajpal et al. (2016)* | PLPC | Short-term cepstral | All-pole autoregressive modelling of Log Filter-Bank |
| *Babu Kalluri et al (2020)* | Functional vector | Long-term statistical | Statistical functions for short-term features |
| *Campbell et al. (2006)* | GMM super vector | Long-term parametric | Parameters of UB-GMM model for short-term features |
| *Dehak et al. (2011)* | I-vector | Long-term parametric | Factorization of GMM supervector |
| *Snyder et al. (2018)* | X-vectors | Neural network | Neural network bottleneck representation |
| *Shon et al. (2017)* | AE embedding | Neural network | Unsupervised representation learning |
| *Brown & Wormald (2017)* | ACC-DIST | Phonotactic | Distance matrix between phoneme acoustics |
| *Najafian & Russell (2020)* | PPRLM | Phonotactic | Sequence and frequency of phoneme usage |

## Classification models

Accent classification models developed in literature from speech can be broadly categorised as utterance-based or phoneme-based models.

## Utterance-based classification

Utterance-based or text-independent models use the complete audio frame for classification and rely on long-term temporal information in the acoustic features. The frame or utterance level acoustic characteristics of speech are then captured by machine learning models (*Soorajkumar et al., 2017*; *Weninger et al., 2019*) for the classification task. However, the choice of machine learning model would depend on the specific task and dataset, and may need to be evaluated using multiple performance metrics and multiple classification scenarios.

PLDA classifiers have been particularly successful for accent and language classification tasks (*Abdurrahman & Zahra, 2021*). SVM has been successfully applied to classify spoken accents from acoustic features of isolated words (*Rizwan & Anderson, 2018*). ELM has been found to be more effective than SVM for the same spoken accent classification from acoustic features of individual spoken words (*Rizwan & Anderson, 2018*).

RNNs are well suited for speech classification tasks due to their feedback catering to the sequential nature of speech (*Adeeba & Hussain, 2019*). CNNs capture the diagonal patterns from the temporal spectrum or Cepstrum. The diagonal patterns represent the transition of frequency characteristics with time and are useful indicators for phoneme articulation as well as speaker characteristics (*Tripathi et al., 2019*).

## Phoneme-based classification

Phoneme-based accent classification models extract only the speech segments representing particular phonemes and then use those particular phonemes and their acoustic features to classify the speech. Both automatic speech recognition and forced alignment using speech transcription are used to segment speech into phonemes (*McAuliffe et al., 2017*).

The phonemes are divided into two major categories: consonants and vowels. Consonants usually have a short duration and are produced by restricting the airflow

**Table 3  Models used for accent classification.**

| Reference | Classifier | Type | Description |
|---|---|---|---|
| *Brown & Wormald (2017)* | SVM | Shallow model | Finds a hyperplane to segregate the data classes |
| *Rizwan & Anderson (2018)* | ELM | Shallow model | Single-layer neural network with only output weights tunable |
| *Abdurrahman & Zahra (2021)* | PLDA | Shallow model | Models data as a mixture of Gaussians |
| *Adeeba & Hussain (2019)* | RNN | Deep learning | Neural network with feedback |
| *Najafian & Russell (2020)* | CNN | Deep learning | A small kernel of neural network sliding across the input |

of the excitation signal within the mouth and are attributed to the place and manner of the restriction. On the other hand, vowels are of longer duration and are formed in the open mouth by creating resonant cavities in the vocal tract by tongue position. This position of the tongue represents the articulatory attributes of the vowel. The resonant cavities in the mouth and the glottis cause suppression of most frequencies and high magnitude for two particular frequencies based on the shape of both resonant cavities. These high-magnitude frequencies of the vowel are also called the formant frequencies and play a major role in accent classification (*Johnson, 2004*). Since vowels and their pronunciations have more significant differences between accents, as compared to consonants, most works have focused on only vowel pronunciations for phoneme-specific classifiers (*Suzuki et al., 2009*).

The two most well-known techniques for phoneme-based accent classification are accent distance (ACCDIST) and parallel phone recognition and language modeling (PPRLM). PPRLM classifiers are based on the hypothesis that speakers from different backgrounds use different sequences and frequencies for phonemes in their speech. Hence, classifiers such as support vector machines (SVMs) use phoneme frequency and sequences to classify the accents (*Chen et al., 2021*).

On the other hand, ACCDIST techniques are based on the hypothesis that people from different backgrounds articulate the vowels differently, and consequently, their vowels have different acoustics. The model classifies spatial distance matrix between the acoustic features for all the vowel instances within the utterance. Mostly, the midpoint MFCC vectors for vowel segments are used as their acoustic features (*Brown & Wormald, 2017*).

Table 3 summarizes the types of machine learning classifiers mostly used in speech accent classification literature.

## COMPARATIVE ANALYSIS

The most common performance measure for evaluating speaker profiling models is accuracy (*Rizwan & Anderson, 2018*; *Singh, Pillay & Jembere, 2020*). Accuracy indicates the ratio of correctly predicted speech samples to the total number of speech samples. However, other performance measures, including precision, recall, and f-score have also been reported and compared for many speaker profiling tasks. Precision for each speaker category in the dataset represents the ratio of true predictions to the number of total predictions for the category. On the other hand, recall indicates the ratio of true predictions to the total

number of samples for each category. F-score encompasses both precision and recall and is computed as the harmonic mean of precision and recall (*Humayun, Yassin & Abas, 2022*). All the category-wise metrics including precision, recall, and f-score can be averaged across all categories to indicate an overall performance measure for the complete dataset. The averaging across categories can be simple or weighted with the number of samples in each class, with weighted averaging capable of highlighting if the classifier is biased towards a particular category. Besides these performance measures, a confusion matrix is usually used to illustrate the number of predictions corresponding to the true samples for the categories. The confusion matrix effectively highlights the common misidentifications between specific categories.

*Brown & Wormald (2017)* has used the AISEB dataset to test forensic accent profiling over similar accents. The dataset consists of speech samples from proximate towns near the Scottish-English border. Experiments were conducted using the ACCDIST-based SVM classifier, with the classifier achieving an accuracy of 86.7% in identifying the four different accents. To test the classification performance for telephone-quality speech, the speech has been degraded by down-sampling and bandpass filtering to mimic the telephone channel. This drops the classification accuracy to 64.4%.

*Weninger et al. (2019)* use a bi-directional long short term memory (LSTM) based deep learning model for text-independent accent classification to improve Mandarin speech recognition. Acoustic features, which have been extracted by sliding windows over a complete audio frame, are used as input to the bi-directional LSTM for temporal modelling. The classifier achieves a low accuracy of 34.1% in identifying the 15 different accents of mandarin from mainland China.

*Jiao et al. (2016b)* merged feed-forward and recurrent neural networks for text-independent, native language identification from the TOEFL dataset for the Native Language Sub Challenge (NLSC). In trying to classify speakers from the 11 different countries, the proposed model achieves an accuracy of 51.92%. Both the DNN and RNN are trained without textual information. The audio samples are segmented into fix-sized 4-second frames for classification, whereby for each 4-second frame, a long-term feature vector is obtained and used as input for the DNN. The long-term feature vector is obtained by computing different statistical functions, including mean and standard deviation, over short-term MFCC features across the frame. On the other hand, the RNN models the temporal series of the short-term features directly. A combination of both RNN and DNN is used as the final classifier. However, the accuracy for the short-term features-based classification model is significantly lower than the I-vector based model on the same dataset, with researchers (*Shivakumar, Chakravarthula & Georgiou, 2016*) reporting an accuracy of 79.93% using I-vector as well as phonetic features for the same NLSC task.

*Najafian & Russell (2020)* use the ABI dataset, which consists of fourteen different accents of British English, to improve and adapt the ASR model. The model uses a PPRLM with SVM for phoneme-based and universal background-Gaussian mixture model for text-independent accent classifications and merge their outputs. It has been shown that the model is able to achieve an accuracy of 84.87%. Prior research by *De Marco & Cox (2013)*

on the same ABI dataset had demonstrated an accuracy of 81.05% by using the acoustic I-vectors only.

*Ge (2015)* combines phoneme-specific and text-independent classification using the Foreign Accented English (FAE) *corpus* classifying seven accents. The proposed model is composed of a combination of phoneme-based and long-term classifiers. Universal background GMM classifier predicts the accent for both, the phonemes and the complete speech utterances. The phonemes which can be recognised with a higher degree of confidence have been selected for phoneme-based weighted classification, with the weight for each classification corresponding to the occurrences in the dataset for that phoneme. Tested with short speech samples, the model is able to achieve an accuracy of 54%. However, the *corpus* consists of 23 different accents from around the world, which are grouped to collapse into seven different categories for classification, and this makes the target accents significantly different and the classification task relatively easier.

A research (*Rizwan & Anderson, 2018*) classifies specific word utterances in speech using ELM with MFCC and its delta features as input, to classify English speech. The research used the TIMIT dataset and achieves 77.88% accuracy for seven native United States accents. *Singh, Pillay & Jembere (2020)* used Speech Accents Archive (SAA) dataset and found that MFCC as short-term features perform the best for accent classification.

Finally, *Ubale et al. (2019)* propose to use attentive pooling in CNNs for accent classification. The model uses neural network for the classification with short term spectral features as input and PLDA with I-vector inputs. The fusion of both classifications achieves 83.32% accuracy over a recent TOEFL dataset managed by the Education Testing Service (ETS). More recently, *Ubale et al. (2019)* applied CNN model directly to a raw audio and merged its output with I-vector classifier. The fusion reported an accuracy of 86.05%. *Kethireddy, Kadiri & Gangashetty (2020)* have also used CNN with the raw audio as input and reported 81.26% accuracy over a subset of CV dataset comprising eight accents.

Table 4 lists benchmark models for speech accent classification with the reported accuracies as well as the strategies and features used. The table demonstrates that until recently, I-vector representations with shallow models, such as support vector machines (SVMs), have been found to perform best for speech accent classification. However, recent studies have shown that using sequences of spectro-temporal features or even raw audio waveforms with CNNs can result in higher classification accuracy. This shift in the use of features and models highlights the evolving nature of the field and the ongoing search for improved methods in speech accent classification. Figure 4 illustrates the ratio for usage of different speech datasets, input features, and classification models by the surveyed studies.

## FUTURE RESEARCH DIRECTIONS

Analyzing the state-of-the-art models and trends in speaker profiling, the following list of key research directions can be identified for future research in speaker profiling.

1. Deep learning-based unsupervised long-term speaker embeddings for profiling
2. Interpretable machine learning for speaker profiling, especially in forensic applications.
3. Exploring the correlation of phoneme articulation features with demographic accents.

**Table 4** Benchmark models for speech accent classification.

| Reference | Dataset | Accents | Accuracy | Features | Classifier | Remarks |
|---|---|---|---|---|---|---|
| *Kethireddy, Kadiri & Gangashetty (2020)* | Common voice | 8 | 81.26% | Raw wave | CNN | CNN applied directly to the raw audio waveform |
| *Ubale et al., (2019)* | TOEFL | 11 | 86.05% | Raw wave, I-vector | CNN, Attentive-pooling, PLDA | CNN applied to raw waveform before weighted global averaging and fusion with PLDA using I-vector |
| *Ubale, Qian & Evanini (2018)* | | 11 | 83.32% | Log Filter-Bank, I-vector | RNN, Attention, CNN, PLDA | Fusion of RNN and CNN applied to Log Filter-Bank features, and PLDA applied to I-vector. |
| *Jiao et al. (2016b)* | | 11 | 51.92% | MFCC, LT vector | RNN, DNN | Fusion of RNN applied to MFCC sequence, and DNN applied to a statistically modelled long-term vector. |
| *Shivakumar, Chakravarthula & Georgiou (2016)* | | 11 | 79.93% | I-vector | PLDA | PLDA applied to I-vector |
| *Rizwan & Anderson (2018)* | TIMIT | 7 | 77.88% | MFCC, deltas | ELM | ELM applied to the combination of MFCC and delta features |
| *Ge (2015)* | FAE | 7 | 54.00% | PLP, PCA, HLDA | UBM-GMM | Universal Background GMM model applied to PLP features compressed using PCA and HLDA |
| *Brown (2018)* | AISEB | 4 | 86.70% | MFCC, ACCDIST | SVM | SVM applied to distance matrix among vowel acoustic features |
| *Najafian & Russell (2020)* | ABI | 4 | 84.87% | PPRLM, I-vector | SVM | Fusion of classification using I-vector and Phonotactic features |
| *De Marco & Cox (2013)* | | 4 | 81.05% | I-vector projections | LDA | LDA used to project I-vectors in lower dimensions before classification |

4. Speaker profiling models for low-resource languages.

Most successful models that utilise long-term acoustic features use I-vectors that are computed *via* statistical measures. Although bottleneck embedding from deep neural networks, termed x-vector, has been used for speaker verification models, deep learning-based long-term modelling has yet to surpass the performance of I-vectors for speaker accent profiling.

Lately, there has been a noticeable increase in research exploring interpretable machine learning and deep learning models for various machine learning tasks. However, the task of speaker profiling research does not have many interpretable models. This is particularly crucial, given the substantial advantages that interpretable models can bring, especially in the context of forensic and sociophonetic applications. Interpretable models can significantly enhance the understanding of the correlation between the articulation features of phonemes and demographic accents. Filling this gap provides a valuable research direction that can help grasp the connections between linguistic nuances and demographic characteristics. Moreover, interpretable results for speaker profiling can be more reliable for forensic communities aimed at criminal conviction which has critical human consequences.

Except for a few studies (*Humayun, Shuja & Abas, 2023*), most of the accent classification models disregard individual phoneme specific time segments, classifying complete utterances. Few works target analysis from individual phoneme time windows. Examining

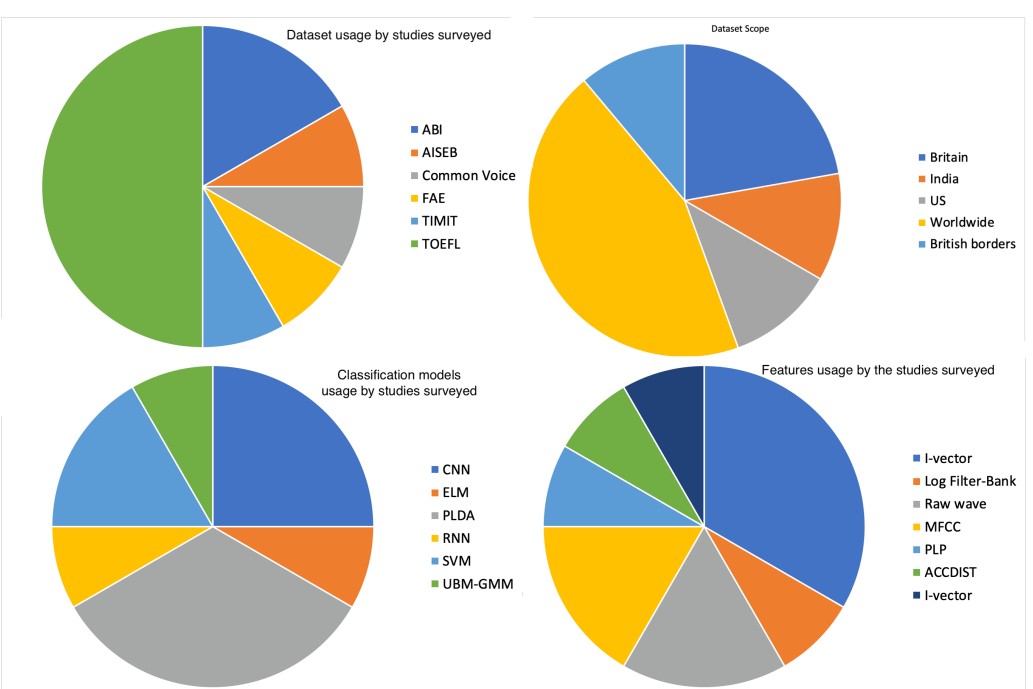

**Figure 4  Usage of speech datasets, input features, and classification models.**

short-term features related to the articulation of a specific phoneme and exploring the variability of these features within a short duration can also significantly aid in enhancing the comprehension of sociophonetic variations. This exploration has potential for future research, particularly in advancing phonetic understanding.

Limited works have analyzed vowel-specific accent classifications. Vowels stand out as the most prominent distinguishing phonemes across various accents, given their usage frequency in speech and their extended durations. Therefore, delving into the articulation characteristics and, consequently, the features unique to vowel-associated speech is an important focus for future research. The specific classification of vowels can not only refine our understanding the classification results but can also enhance interpretable and reliable speaker profiling.

Machine learning-based research demands extensive datasets and substantial resources, often directing its attention predominantly to high-resource languages, such as English, while neglecting low-resource languages, particularly those indigenous to areas with limited resources. Despite the presence of accent variations in many of these low-resource languages, minimal research has been dedicated to speaker profiling within this context. Consequently, there is a significant research potential in exploring low-resource languages for speaker profiling.

Cross-lingual transfer learning capability can also be explored in the future, particularly for low resource languages. Accent or speaker profiling models can be tailored for low-resource languages by adapting through the fine-tuning of models initially trained on high-resource languages. This approach offers a viable solution to address the scarcity of

labeled datasets in low-resource languages by transferring the knowledge gained in speaker profiling from high-resource languages.

## CONCLUSIONS

Speaker profiling ranges from identifying the physical body parameters of speakers to identifying their social traits, geographic origin, and native language. Estimation of the social background of speakers has multiple applications, including forensic investigations and improving speech recognition models.

This article has reviewed the state-of-the-art in speaker profiling from speech accents. The speech features, generally used as input for speaker accent classification, have been presented. Short-term acoustic features are mostly modelled as long-term vectors for complete speech utterances to classify the accents. Traditional machine learning classifiers using I-vector as long-term acoustic representation were the most successful for speaker profiling tasks. However, deep learning models have recently outperformed conventional classification methods by using recurrent and convolutional neural networks on the sequence of short-term spectral features or even directly on the raw audio waveform.

Most notable among the speech datasets for English is the TOEFL English speech dataset by ETS, containing speech samples from candidates worldwide and has been used for native language identification tasks. The highest accuracy of 86.05% has been reported on the ETS dataset by applying a bank of CNNs directly to the raw audio waveforms of speech.

### Funding
The authors received no funding for this work.

### Competing Interests
Junaid Shuja is an Academic Editor for PeerJ.

### Author Contributions
- Mohammad Ali Humayun conceived and designed the experiments, performed the experiments, analyzed the data, performed the computation work, prepared figures and/or tables, authored or reviewed drafts of the article, and approved the final draft.
- Junaid Shuja conceived and designed the experiments, performed the experiments, analyzed the data, performed the computation work, prepared figures and/or tables, authored or reviewed drafts of the article, and approved the final draft.
- Pg Emeroylariffion Abas conceived and designed the experiments, performed the experiments, analyzed the data, performed the computation work, prepared figures and/or tables, authored or reviewed drafts of the article, and approved the final draft.

### Data Availability
This is a literature review.

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
