# Peer review of "A review of social background profiling of speakers from speech accents"

_PeerJ Computer Science, doi:10.7717/peerj-cs.1984_

## Round 0.1 · original submission · Major Revisions

Revised paper based on reviewers' suggestions.

**Language Note:** The review process has identified that the English language must be improved. PeerJ can provide language editing services - please contact us at copyediting@peerj.com for pricing (be sure to provide your manuscript number and title). Alternatively, you should make your own arrangements to improve the language quality and provide details in your response letter. – PeerJ Staff

Reviewer 1 ·

Basic reporting

No comments

Experimental design

No comments

Validity of the findings

In the paper titled “A review of social background profiling of speakers from speech accents” authors review the speech datasets for background profiling of speakers. This paper offers a survey of recent research on automatic speech accent classification, focusing on its applications in social background profiling, particularly in areas such as speech forensics and improving speech recognition accuracy. The analysis covers datasets, speech features, and classification models used in accent classification tasks, aiming to provide a comprehensive overview and comparative analysis of performance measures. The study identifies research gaps in speech accent classification, offering valuable insights for researchers seeking to advance the field. My concerns and suggestions as below
• The introduction is not well written. There are some doubts on the applicability of social profiling from speaker accents as stated by the authors themselves. Consider line 46-48 stating that emotional characteristics are also present in speech. Do the listed studies focus on the emotional state of speaker for social profiling? Are other features of the speaker such as education, Socioeconomic status, emotional state present in the data sets? Relevant to social background profiling?
• As the article domain is interdisciplinary, are there disciplines other than forensics where speaker profiling can be applied?
• The introduction does not identify the gap in existing literature. If any reviews have been published, how does this review distinguish?
• The survey methods lack details. What was n = the initial number of articles from the keyword search? What happened to n after applying filters? What was the last value of n?
• There should be labels in figures. Use labels or full form.
• Are there any references for figure 2 and 3?
• The research directions should be itemized.

Additional comments

No comments

Cite this review as

Reviewer 2 ·

Basic reporting

This paper surveys recent research in speech accent classification and analyses the datasets, speech features, and classification models used for the classification tasks. The aim is to provide a comprehensive overview of recent research related to speech accent classification and to present a comparative analysis of the achieved performance measures. Comprehensive descriptions of the datasets, speech features, and classification models used in recent research related to speech accent classification have been presented, with a comparative analysis of the performance measures of the different methods. This analysis provides insights into the strengths and weaknesses of the different methods. Subsequently, research gaps in speech accent classification have been identified. The idea for the review is novel but the paper needs major revisions:
The start of the introduction is poor, as a novice reader will face difficulties while grasping the information from the introduction. It should be modified according to the hourglass fashion. First, the overall field should be discussed. Later on, it should slightly move to the original contribution with a proper flow.

Experimental design

The author should write the article's selection criteria with a more specific selection. It seems generic criteria which may lead to huge exclusion or inclusion causing a loss of critical information in the literature selection.
This seems to be a systematic review, then why crucial steps are missing, such as protocol selection, PRISMA or Kitchenham, a complete flow of step-by-step inclusion or exclusion, total number of articles extracted, removed articles based on various factors such as full articles accessibility, etc.

Validity of the findings

Why results are missing based on factors such as year-wise publications, trends, etc.? It would be better to provide the results with pie charts etc.
The information within the tables is well presented.
Overall the article needs critical results figures of reviewed articles based on some good parameters that are normally used in systematic reviews.

Cite this review as

---

## Round 0.2 · accepted · Accept

The paper is acceptable now.

Reviewer 1 ·

Basic reporting

The authors have addressed all my previous comments. I now deem the paper acceptable for publication.

Experimental design

The authors have addressed all my previous comments. I now deem the paper acceptable for publication.

Validity of the findings

The authors have addressed all my previous comments. I now deem the paper acceptable for publication.

Additional comments

The authors have addressed all my previous comments. I now deem the paper acceptable for publication.

Cite this review as

Reviewer 2 ·

Basic reporting

The revised manuscript looks ready to be published

Experimental design

The requested changes and suggestions are incorporated

Validity of the findings

The major revision I suggested looks incorporated.

Additional comments

This revision incorporates all my comments.

Cite this review as